# DEEP ATTENTIVE VARIATIONAL INFERENCE

**Ifigeneia Apostolopoulou**[1,2]**, Ian Char**[1,2]**, Elan Rosenfeld**[1] **& Artur Dubrawski**[1,2]
[1]Machine Learning Department & [2]AutonLab, Carnegie Mellon University
Correspondence to: `iapostol@andrew.cmu.edu, ifiaposto@gmail.com`

## ABSTRACT

Stochastic Variational Inference is a powerful framework for learning large-scale probabilistic latent variable models. However, typical assumptions on the factorization or independence of the latent variables can substantially restrict its capacity for inference and generative modeling. A major line of active research aims at building more expressive variational models by designing deep hierarchies of interdependent latent variables. Although these models exhibit superior performance and enable richer latent representations, we show that they incur diminishing returns: adding more stochastic layers to an already very deep model yields small predictive improvement while substantially increasing the inference and training time. Moreover, the architecture for this class of models favors proximate interactions among the latent variables between neighboring layers when designing the conditioning factors of the involved distributions. This is the first work that proposes attention mechanisms to build more expressive variational distributions in deep probabilistic models by explicitly modeling both nearby and distant interactions in the latent space. Specifically, we propose deep attentive variational autoencoder and test it on a variety of established datasets. We show it achieves state-of-the-art log-likelihoods while using fewer latent layers and requiring less training time than existing models. The proposed holistic inference reduces computational footprint by alleviating the need for deep hierarchies. Project code: `https://github.com/ifiaposto/Deep_Attentive_VI`

## 1 INTRODUCTION

A core line of research in both supervised and unsupervised learning relies on deep probabilistic models. This class of models uses deep neural networks to model distributions that express hypotheses about the way in which the data have been generated. Such architectures are preferred due to their capacity to express complex, non-linear relationships between the random variables of interest while enabling tractable inference and sampling. Latent variable probabilistic models augment the set of the observed variables with auxiliary latent variables. They are characterized by a posterior distribution over the latent variables, one which is generally intractable and typically approximated by closed-form alternatives. They provide an explicit parametric specification of the joint distributions over the expanded random variable space, while the distribution of the observed variables is computed by marginalizing over the latent variables. The Variational Autoencoder (VAE) (Kingma & Welling, 2014) belongs to this model category. The VAE uses neural networks for learning the parametrization of both the inference network (which defines the posterior distribution of the latent variables) and the generative network (which defines the prior distribution of the latent variables and the conditional data distribution given the latent variables). Their parameters are jointly learned via stochastic variational inference (Paisley et al., 2012; Hoffman et al., 2013).

Early VAE architectures (Rezende et al., 2014) make strong assumptions about the posterior distribution—specifically, it is standard to assume that the posterior is approximately factorial. Since then, research has progressed on learning more expressive latent variable models. For example, Rezende & Mohamed (2015); Kingma et al. (2016); Chen et al. (2017) aim at modeling more complex posterior distributions, constructed with autoregressive layers. Theoretical research focuses on deriving tighter bounds (Burda et al., 2016; Li & Turner, 2016; Masrani et al., 2019) or building more informative latent variables by mitigating posterior collapse (Razavi et al., 2019a; Lucas et al., 2019). Sinha & Dieng (2021) improve generalization by enforcing regularization in the latent space

for semantics-preserving transformations of the data. Taking a different approach, hierarchical VAEs (Gulrajani et al., 2017; Sønderby et al., 2016; Maaløe et al., 2019; Vahdat & Kautz, 2020; Child, 2020) leverage increasingly deep and interdependent layers of latent variables, similar to how subsequent layers in a discriminative network are believed to learn increasingly abstract representations of the data. These architectures exhibit superior generative and reconstructive capabilities since they allow for modeling of much richer structures in the latent space.

Previous work overlooks the effect of long-range correlations among the latent variables. In this work, we propose to restructure common hierarchical, convolutional VAE architectures in order to increase the receptive field of the variational distributions. We first provide experimental evidence that conditional dependencies in deep probabilistic hierarchies may be implicitly disregarded by current models. Subsequently, we propose a decomposed, attention-guided scheme that allows a long-range flow of both the latent and the observed information both across different, potentially far apart, stochastic layers and within the same layer and we investigate the importance of each proposed change through extensive ablation studies. Finally, we demonstrate that our model is both computationally more economical and can attain state-of-the-art performance across a diverse set of benchmark datasets.

## 2 PROPOSED MODEL

### 2.1 DEEP VARIATIONAL INFERENCE

A latent variable generative model defines the joint distribution of a set of observed variables, $\boldsymbol{x} \in \mathbb{R}^D$, and auxiliary latent variables, $\boldsymbol{z}$, coming from a prior distribution $p(\boldsymbol{z})$. To perform inference, the marginal likelihood of the distribution of interest, $p(\boldsymbol{x})$, can be computed by integrating out the latent variables:

$$p(\boldsymbol{x}) = \int p(\boldsymbol{x}, \boldsymbol{z}) \, d\boldsymbol{z}. \tag{1}$$

Since this integration is generally intractable, a lower bound on the marginal likelihood is maximized instead. This is done by introducing an approximate posterior distribution $q(\boldsymbol{z} \mid \boldsymbol{x})$ and applying Jensen's inequality:

$$\log p(\boldsymbol{x}) = \log \int p(\boldsymbol{x}, \boldsymbol{z}) \, d\boldsymbol{z} = \log \int \frac{q(\boldsymbol{z} \mid \boldsymbol{x})}{q(\boldsymbol{z} \mid \boldsymbol{x})} p(\boldsymbol{x}, \boldsymbol{z}) \, d\boldsymbol{z} \geq \int q(\boldsymbol{z} \mid \boldsymbol{x}) \log \left[ \frac{p(\boldsymbol{x} \mid \boldsymbol{z}) p(\boldsymbol{z})}{q(\boldsymbol{z} \mid \boldsymbol{x})} \right] \, d\boldsymbol{z}$$

$$\implies \log p(\boldsymbol{x}) \geq \mathbb{E}_{q(\boldsymbol{z} \mid \boldsymbol{x})}[\log p(\boldsymbol{x} \mid \boldsymbol{z})] - D_{KL}(q(\boldsymbol{z} \mid \boldsymbol{x}) \parallel p(\boldsymbol{z})), \tag{2}$$

where $\boldsymbol{\theta}$, $\boldsymbol{\phi}$ parameterize $p(\boldsymbol{x}, \boldsymbol{z}; \boldsymbol{\theta})$ and $q(\boldsymbol{z} \mid \boldsymbol{x}; \boldsymbol{\phi})$ respectively. For ease of notation, we omit $\boldsymbol{\theta}, \boldsymbol{\phi}$ in the derivations. This objective is called the Evidence Lower BOund (ELBO) and can be optimized efficiently for continuous $\boldsymbol{z}$ via stochastic gradient descent (Kingma & Welling, 2014; Rezende et al., 2014).

For ease of implementation, it is common to assume that both $q(\boldsymbol{z} \mid \boldsymbol{z})$ and $p(\boldsymbol{z})$ are fully factorized Gaussian distributions. However, this assumption may be too limiting in cases of complex underlying distributions. To compensate for this modeling constraint, many works focus on stacking and improving the stability of multiple layers of stochastic latent features which are partitioned in groups such that $\boldsymbol{z} = \{\boldsymbol{z}_1, \boldsymbol{z}_2, \ldots, \boldsymbol{z}_L\}$, where $L$ is the number of such groups (Rezende et al., 2014; Gulrajani et al., 2017; Kingma et al., 2016; Sønderby et al., 2016; Maaløe et al., 2019; Vahdat & Kautz, 2020; Child, 2020). Our work builds on architectures of bidirectional inference with a deterministic bottom-up pass. The schematic diagram of a stochastic layer in such a deep variational model is depicted in Figure 1. In a bidirectional inference architecture with a deterministic bottom-up pass (left part in Figure 1a), posterior sampling is preceded by a sequence of non-linear transformations, $T_l^q$, of the evidence, $\boldsymbol{x}$, i.e., $\boldsymbol{h}_l = T_l^q(\boldsymbol{h}_{l+1})$, with $\boldsymbol{h}_{L+1} = \boldsymbol{x}$. The inference $q(\boldsymbol{z} \mid \boldsymbol{x})$ and generative $p(\boldsymbol{z})$ network decompose in an identical topological ordering: $q(\boldsymbol{z} \mid \boldsymbol{x}) = \prod_l q(\boldsymbol{z}_l \mid \boldsymbol{x}, \ \boldsymbol{z}_{<l})$ and $p(\boldsymbol{z}) = \prod_l p(\boldsymbol{z}_l \mid \boldsymbol{z}_{<l})$. The top-down pass (right part in Figure 1a) generates the posterior samples $\boldsymbol{z}$ that feed the conditional data distribution $p(\boldsymbol{x} \mid \boldsymbol{z})$.

### 2.2 MOTIVATION

Kingma et al. (2016) proposed a strongly connected directed probabilistic graphical model for the generative and inference network, so that each variable depends on all the previous in the hierarchy:

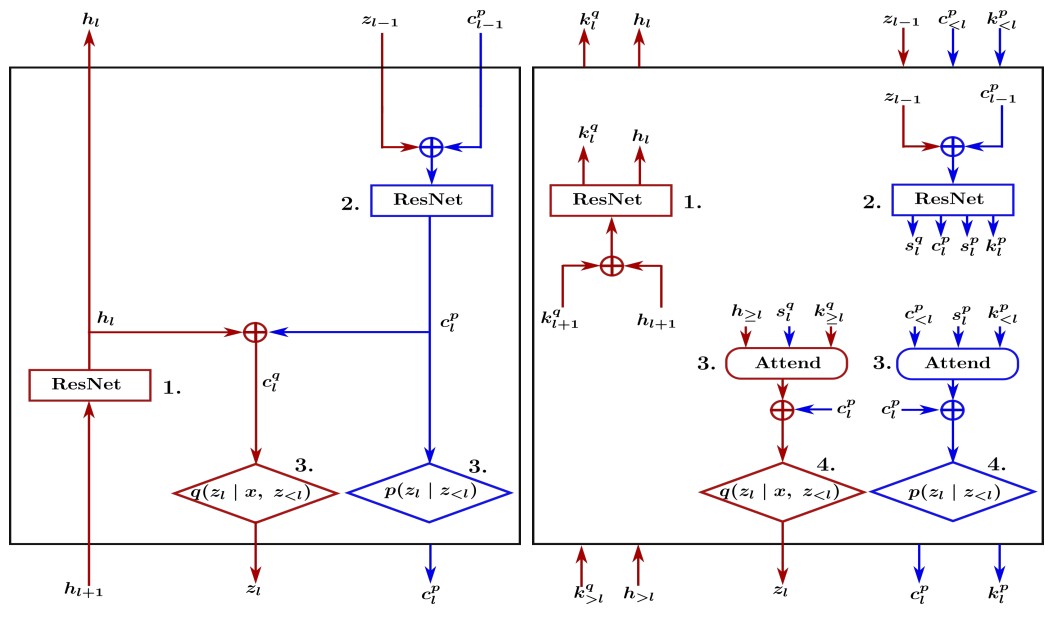

Figure 1: **Computational graph of the $l$-th variational layer in a hierarchy.** The numbers denote the order of the computations. The inference path forms the posterior $q(z_l \mid x, z_{<l})$ given the condition $c_l^q$. The generative path forms the prior $p(z_l \mid z_{<l})$ given the condition $c_l^p$. $c_l^p$ represents context information that consists of both deterministic features and latent samples with $c_0^p$ being a constant. $h_l$ represents hidden features of the data with $h_{L+1} \equiv x$. Multiple such blocks are stacked. $p(x \mid z)$ receives the sample $z_L$ and the context $c_L^p$ of the last layer. The $\oplus$ symbol denotes a module responsible for combining two streams of features. In Figure 1a, the layer is connected only with the adjacent layers in the hierarchy. Latent information of earlier layers $z_{<l-1}$ is carried through $c_{l-1}^p$. In Figure 1b, the layer is connected with all the layers below (above) at the bottom-up pass (top-down pass). The ResNet transformations are extended to produce key $k$ and query $s$ feature maps. The generative network queries the layers of the generative network (inference network) above (below) to identify the most relevant conditioning factors of the prior (posterior) according to attention scores computed by the attention modules (see Figure 2).

$p(z_l \mid z_{<l})$. Similarly, for the inference model: $q(z_l \mid x, z_{<l})$. This is in contrast to other works (Sønderby et al., 2016) that consider statistical dependencies between successive layers only, i.e., $p(z_l \mid z_{l-1})$. Maaløe et al. (2019) also highlight the importance of this modification. The long-range conditional dependencies are implicitly enforced via deterministic features that are mixed with the latent variables and propagated through the hierarchy (see feature $c_l^p$ in Figure 1a).

State-of-the-art models (Vahdat & Kautz, 2020; Child, 2020) leverage this fully-connected factorization and rely on the increased depth to improve performance and deliver results comparable to that of autoregressive models (Salimans et al., 2017). However, by construction, very deep VAE architectures favor only proximate dependencies in the latent space, limiting long-range conditional dependencies between $z_l$ and $z_{<l-1}$ as depth increases. This means that in practice the network may no longer respect the factorization of the variational distributions $p(z) = \prod_l p(z_l \mid z_{<l})$ and $q(z \mid x) = \prod_l q(z \mid x, z_{<l})$, leading to sub-optimal performance.

Table 1: $-\log p(x)$ for **varying depth $L$ (bits/dim)**.

| Depth $(L)$ | bits/dim $\downarrow$ | $\Delta(\cdot)\%$ |
|---|---|---|
| 2 | 3.50 | − |
| 4 | 3.26 | −6.8 |
| 8 | 3.06 | −6.1 |
| 16 | 2.96 | −3.2 |
| 30 | 2.91 | −1.7 |

Table 1 reports the absolute and relative decrease in the negative log-likelihood (in bits per dimension) as one increases the number of stochastic layers in an NVAE (Vahdat & Kautz, 2020). We observe that the predictive gains diminish as depth increases. We hypothesize that this may be because the effect of the latent variables of earlier layers diminishes as the context feature $c_l^p$ traverses the hierarchy and is updated with latent information from subsequent layers.

In this work, we improve the flexibility of the prior $p(\boldsymbol{z})$ and posterior $q(\boldsymbol{z} \mid \boldsymbol{x})$ by designing more informative representations for the conditioning factors of the conditional distributions $p(\boldsymbol{z}_l \mid \boldsymbol{z}_{<l})$ and $q(\boldsymbol{z}_l \mid \boldsymbol{x}, \boldsymbol{z}_{<l})$. We do this by designing a hierarchy of densely connected stochastic layers that learn to attend to latent and observed information most critical to inference. Figure 1b is a graphical illustration of the proposed model.

## 2.3 Depth-Wise Attention

We first introduce depth-wise attention. This is the technical tool that allows our model to realize the strong couplings presented in Figure 1b and motivated in Section 2.2. The problem can be formulated as follows:

*Given a sequence $\boldsymbol{c}_{<l} = \{\boldsymbol{c}_m\}_{m=1}^{l-1}$ of $l-1$ contexts, we need to construct a feature $\hat{\boldsymbol{c}}_l$ that summarizes the information in $\boldsymbol{c}_{<l}$ that is most critical for a given task. $\hat{\boldsymbol{c}}_l$ and $\boldsymbol{c}_m$ are features of the same dimensionality: $\hat{\boldsymbol{c}}_l \in \mathbb{R}^{H \times W \times C}$, and $\boldsymbol{c}_m \in \mathbb{R}^{H \times W \times C}$.*

In our framework, this task is the construction of either prior (Section 2.4) or posterior (Section 2.5) beliefs of a variational layer in a deep VAE. Therefore, our architecture must be able to handle long context sequences of large dimensions $H$ and $W$.

The task is characterized by a query feature $\boldsymbol{s}_l \in \mathbb{R}^{H \times W \times Q}$ of dimensionality $Q$ with $Q \ll C$. Similarly, $\boldsymbol{c}_m$ is represented by a key feature $\boldsymbol{k}_m \in \mathbb{R}^{H \times W \times Q}$. To reduce computational requirements, we treat each pixel independently

Figure 2: **Attend**$(\boldsymbol{c}_{<l},\ \boldsymbol{s}_l,\ \boldsymbol{k}_{<l})-$ **a depth-wise attention block.** $\boldsymbol{c}_{<l}, \boldsymbol{k}_{<l}$ are the sequences of $l-1$ contexts and corresponding keys with $C$ and $Q$ feature maps accordingly, while $\boldsymbol{s}_l$ is the query feature. The multiplication is applied to the inner matrix dimensions. The normalization of the softmax is applied to the last dimension, treating each pixel independently from the others. $\boldsymbol{\alpha}_{<l} = \{\boldsymbol{\alpha}_{m \to l}\}_{m=1}^{l-1}$ are the attention scores.

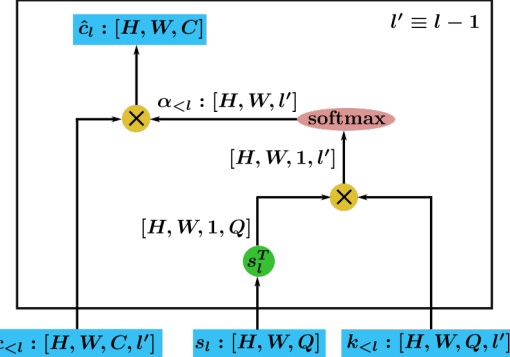

from the rest. This can equivalently be interpreted as concurrent processing of $H \times W$ independent sequences of $C$-dimensional features. The feature maps $\hat{\boldsymbol{c}}_l(i,j) \in \mathbb{R}^C$ for pixels $(i,j)$, with $1 \le i \le H, 1 \le j \le W$, are computed in parallel. $\hat{\boldsymbol{c}}_l(i,j)$ depends only on pixel instances in $\boldsymbol{c}_{<l}$ at the same location $(i,j)$, i.e, $\boldsymbol{c}_m(i,j)$ with $m < l$, and is given by:

$$\hat{\boldsymbol{c}}_l(i,j) = \sum_{m<l} \boldsymbol{\alpha}_{m \to l}(i,j) \boldsymbol{c}_m(i,j), \tag{3}$$

$$\boldsymbol{\alpha}_{m \to l}(i,j) = \frac{\exp(\boldsymbol{s}_l^T(i,j)\boldsymbol{k}_m(i,j))}{\sum_{m<l} \exp(\boldsymbol{s}_l^T(i,j)\boldsymbol{k}_m(i,j))}. \tag{4}$$

In words, feature $\boldsymbol{s}_l(i,j) \in \mathbb{R}^Q$ queries the significance of feature $\boldsymbol{c}_m(i,j) \in \mathbb{R}^C$, represented by $\boldsymbol{k}_m(i,j) \in \mathbb{R}^Q$, to form $\hat{\boldsymbol{c}}_l(i,j) \in \mathbb{R}^C$. $\boldsymbol{\alpha}_{m \to l}(i,j) \in \mathbb{R}$, is the resulting relevance metric of the $m-$th term, with $m < l$, at pixel $(i,j)$. The overall procedure is denoted as $\hat{\boldsymbol{c}} = \text{Attend}(\boldsymbol{c}_{<l},\ \boldsymbol{s}_l,\ \boldsymbol{k}_{<l})$, and is illustrated in Figure 2.

Finally, to improve training of models with very long sequences, we adopt the following variant of the normalization scheme proposed by Chen et al. (2020):

$$\boldsymbol{c}_{<l} \leftarrow \boldsymbol{c}_{<l} + \text{Gelu}(\text{LayerNorm}(\boldsymbol{c}_{<l})), \tag{5}$$

$$\hat{\boldsymbol{c}}_l \leftarrow \hat{\boldsymbol{c}}_l + \text{Gelu}(\text{LayerNorm}(\hat{\boldsymbol{c}}_l)), \tag{6}$$

where Gelu is the GELU non-linearity (Hendrycks & Gimpel, 2016) and LayerNorm a layer normalization operation (Ba et al., 2016).

## 2.4 Generative Model

As shown in Figure 1a, the conditioning factor of the prior distribution at variational layer $l$ is represented by context feature $\boldsymbol{c}_l^p \in \mathbb{R}^{H \times W \times C}$. A convolution is applied on $\boldsymbol{c}_l^p$ to obtain parameters

$\boldsymbol{\theta}$ defining the prior. $\boldsymbol{c}_l^p$ is a non-linear transformation of the immediately previous latent information $\boldsymbol{z}_{l-1}$ and prior context $\boldsymbol{c}_{<l-1}$ containing latent information from distant layers $\boldsymbol{z}_{<l-1}$, such that $\boldsymbol{c}_l^p = T_l^p(\boldsymbol{z}_{l-1} \oplus \boldsymbol{c}_{l-1}^p)$. $T_l^p(\cdot)$ is typically implemented as a cascade of ResNet cells (He et al., 2016a;b) and corresponds to the blue ResNet module in Figure 1a. The operator $\oplus$ combines information from two branches in the network (e.g. by summation or concatenation). $\boldsymbol{z}_{l-1}$, $\boldsymbol{c}_{l-1}^p$ are passed in from the previous layer.

Due to the locality of the architecture, the signal from $\boldsymbol{c}_l^p$ may be dominated by that of $\boldsymbol{z}_{l-1}$. To prevent this, we adopt direct connections between each pair of stochastic layers, as shown in Figure 1b. That is, variational layer $l$ has direct access to prior context of all previous layers $\boldsymbol{c}_{<l}^p$ accompanied by keys $\boldsymbol{k}_{<l}^p$. This means each variational layer can actively determine the most important latent contexts when evaluating its prior beliefs. During training, the context, query, and key are jointly learned:

$$[\boldsymbol{c}_l^p, \ \boldsymbol{s}_l^p, \ \boldsymbol{k}_l^p] \leftarrow T_l^p(\boldsymbol{z}_{l-1} \oplus \boldsymbol{c}_{l-1}^p). \tag{7}$$

We initially let variational layer $l$ rely on nearby dependencies captured by $\boldsymbol{c}_l^p$. During training, the prior is progressively updated with the holistic context $\hat{\boldsymbol{c}}_l^p$ via a residual connection (Wang et al., 2018; Bachlechner et al., 2021):

$$\hat{\boldsymbol{c}}_l^p \leftarrow \text{Attend}(\boldsymbol{c}_{<l}^p, \ \boldsymbol{s}_l^p, \ \boldsymbol{k}_{<l}^p), \tag{8}$$

$$\hat{\boldsymbol{c}}_l^p \leftarrow \boldsymbol{c}_l^p + \gamma_l^p \hat{\boldsymbol{c}}_l^p, \tag{9}$$

where $\gamma_l^p \in \mathbb{R}$ is a learnable scalar parameter initialized to zero, $\boldsymbol{c}_{<l}^p = \{\boldsymbol{c}_m^p\}_{m=1}^{l-1}$ with $\boldsymbol{c}_m^p \in \mathbb{R}^{H \times W \times C}$, $\boldsymbol{s}_l^p \in \mathbb{R}^{H \times W \times Q}$, $\boldsymbol{k}_{<l}^p = \{\boldsymbol{k}_m^p\}_{m=1}^{l-1}$ with $\boldsymbol{k}_m^p \in \mathbb{R}^{H \times W \times Q}$, and $Q \ll C$.

## 2.5 Inference Model

As shown in Figure 1a, the conditioning context $\boldsymbol{c}_l^q$ of the posterior distribution results from combining deterministic factor $\boldsymbol{h}_l$ and stochastic factor $\boldsymbol{c}_l^p$ provided by the decoder: $\boldsymbol{c}_l^q = \boldsymbol{h}_l \oplus \boldsymbol{c}_l^p$. To improve inference, we let layer's $l$ encoder use both its own $\boldsymbol{h}_l$ and all subsequent hidden representations $\boldsymbol{h}_{>l}$, as shown in Figure 1b. This modification can also be viewed as skip connections which selectively connect each layer with representations closer to the data helping deep signal propagation. As in the generative model, the bottom-up path is extended to emit low dimensional key features $\boldsymbol{k}_l^q$ which represent hidden features $\boldsymbol{h}_l$:

$$[\boldsymbol{h}_l, \ \boldsymbol{k}_l^q] \leftarrow T_l^q(\boldsymbol{h}_{l+1} \oplus \boldsymbol{k}_{l+1}^q). \tag{10}$$

Prior works (Sønderby et al., 2016; Vahdat & Kautz, 2020) have sought to mitigate against exploding $D_{KL}$ in Equation 2 by using parametric coordination between the prior and posterior distributions. Motivated by this insight, we seek to establish further communication between them. We accomplish this by allowing the generative model to choose the most explanatory features in $\boldsymbol{h}_{\geq l}$ by generating query feature $\boldsymbol{s}_l^q$. Finally, the holistic conditioning factor for the posterior is:

$$\hat{\boldsymbol{c}}_l^q \leftarrow \text{Attend}(\boldsymbol{h}_{\geq l}, \ \boldsymbol{s}_l^q, \ \boldsymbol{k}_{\geq l}^q). \tag{11}$$

The detailed description is deferred to Appendix A.1. Inference on the model is also summarized in Appendix A.3.

## 2.6 Spatially attentive variational layers

In Sections 2.4 and 2.5, we allowed the perceptive field of each pixel at layer $l$ to encompass only pixels at the same locations in earlier layers during inference. In order to fully capture distant dependencies in the latent space, we still need to coordinate portions of information in the spatial domain. To increase the computational efficiency of our model, this occurs at a second stage. Conceptually, this two-step procedure resembles the factorized attention of efficient transformers (Child et al., 2019), albeit deep variational models are inherently amenable to a different, inter-layer and intra-layer, decomposition. This observation decreases the time and memory complexity of the long-range operations for computing the attention scores from $\mathcal{O}(L^3 \times W^2 \times H^2)$ to $\mathcal{O}(L^2 + L \times W^2 \times H^2)$, while still providing inference with holistic information. To accomplish this, we interleave spatially non-local blocks (Wang et al., 2018) with the convolutions in the residual cells of the ResNet modules in Figure 1, as shown in Figure 3 in Appendix. The exact formulation is similar to that of Wang et al. (2018) and is provided in Appendix A.2.

## 3 EXPERIMENTAL STUDIES

We conduct three series of experiments. In the first set of experiments (Section 3.1.1), we apply the attentive variational path proposed in this paper on VAEs that are trained on two datasets of binary images: the dynamically binarized MNIST and OMNIGLOT. In Section 3.1.2, we investigate the effectiveness of the proposed techniques on large-scale latent spaces that are used for generating the CIFAR-10 natural images. Qualitative results are provided in Appendices E (plot of KL divergence per layer), F (visualization of attention patterns), and G (novel samples and image reconstructions). Finally, in Section 3.2 we conduct an ablation study and report the benefits of each proposed attention module separately. All hyperparameters for experiments are available in Appendix B. We note here that we choose to not use spectral regularization, and we find this does not compromise training stability of our model (in contrast to results reported by Vahdat & Kautz (2020)). The fact that our model is able to achieve better performance without this regularization may indicate that the attention operations help stabilize training.

### 3.1 MAIN QUANTITATIVE RESULTS

#### 3.1.1 APPLICATION ON BINARY IMAGES

**Datasets.** We evaluate the models on two benchmark datasets: MNIST (LeCun et al., 1998), a dataset of $28 \times 28$ images of handwritten digits, and OMNIGLOT (Lake et al., 2013), an alphabet recognition dataset of $28 \times 28$ images. For convenience, we add two zero pixels to each border of the training images. In both cases, the observations are dynamically binarized by being resampled from the normalized real values using a Bernoulli distribution after each epoch, as suggested by Burda et al. (2016), which prevents over-fitting. We use the standard splits of MNIST into 60,000 training and 10,000 test examples, and of OMNIGLOT into 24,345 training and 8,070 test examples.

**Set-up.** For both datasets, we use a hierarchy of $L = 15$ variational layers. We use a Bernoulli distribution in the image decoder. In the generative model a non-local block is inserted at the end of the module responsible for combining the context with the latent sample provided by the previous layer. In the inference layer, a non-local block is inserted at the end of the module responsible for concatenating the deterministic, data-dependent features of the bottom-up pass and the context containing latent information that is provided by the generative model. Non-local blocks are also inserted at the post-processing cells, at the end of the variational hierarchy and right before the layers that implement the data distribution (see Appendix B for details). MNIST images are downsampled twice before being passed to the hierarchy leading to latent spaces of spatial dimension $8 \times 8$ while the OMNIGLOT images are downsampled only once resulting to $16 \times 16$ latent spaces.

**Results.** Table 2 reports the estimated marginal likelihood of our model along with the performance achieved by state-of-the-art models. We observe that in both datasets, holistic variational inference consistently improves performance. More importantly, our model outperforms models that increase the flexibility of the variational posterior by making use of normalizing flows (Kingma et al., 2016) or the data distribution by making use of autoregressive decoders (Chen et al., 2017). The use of autoregressive layers is a model refinement orthogonal to ours and it could potentially further improve the results presented in Table 2. The architecture closest to ours is NVAE (Vahdat & Kautz, 2020), and the performance gap can be solely attributed to long-range inference operations described in Sections 2.4, 2.5, and 2.6.

#### 3.1.2 APPLICATION ON NATURAL IMAGES

**Dataset.** CIFAR-10 is a dataset of $32 \times 32$ natural images. The raw pixel values are first scaled to $[-1, 1]$.

**Set-up.** For CIFAR-10, we use a hierarchy of $L = 16$ variational layers. We use a mixture of discretized Logistic distributions (Salimans et al., 2017) for the data distribution. We use the spatially non-local residual cells of Figure 3 in both the generative and inference network of each layer. Non-local blocks are also inserted at the pre-processing blocks, at the beginning of the bottom-up pass (see Appendix B for details). CIFAR-10 images are downsampled only once resulting to $16 \times 16$ latent spaces. We note that it helps optimization if we bound the log of the prior standard deviation such that $\log \sigma_p \geq -1.0$, yielding less confident prior assumptions. We also empirically find that

Table 2: **Dynamically binarized MNIST** (Burda et al., 2016) **and OMNIGLOT** (Lake et al., 2013) **performance on the test set.** All models except for IWAE are trained with a single importance sample. IWAE is trained with 50 importance samples. The marginal loglikelihood is estimated with 500 importance samples. Attentive VAE outperforms all state-of-the-art VAEs with or without autoregressive components.

| Model | $\log p(\boldsymbol{x}) \geq \uparrow$ | |
|---|---|---|
| | MNIST | OMNIGLOT |
| **Attentive VAE (ours)** | $\mathbf{-77.63}$ | $\mathbf{-89.50}$ |
| NVAE (Vahdat & Kautz, 2020) | $-78.01$ | $-90.18$ |
| MAE (Ma et al., 2019) | $-77.98$ | $-$ |
| BIVA (Maaløe et al., 2019) | $-78.41$ | $-93.54$ |
| PixelVAE++ (Sadeghi et al., 2019) | $-78.00$ | $-$ |
| DVAE++ (Vahdat et al., 2018) | $-78.49$ | $-$ |
| VampPrior (Tomczak & Welling, 2018) | $-78.45$ | $-89.76$ |
| SA-VAE (Kim et al., 2018) | $-$ | $-90.05$ |
| Lossy VAE (Chen et al., 2017) | $-78.53$ | $-89.83$ |
| Ladder VAE (Sønderby et al., 2016) | $-81.74$ | $-102.11$ |
| IAF-VAE (Kingma et al., 2016) | $-79.10$ | $-$ |
| DVAE (Rolfe, 2017) | $-81.01$ | $-97.43$ |
| Conv DRAW (Gregor et al., 2016) | $-$ | $-91.00$ |
| IWAE (Burda et al., 2016) | $-82.90$ | $-103.38$ |

adding Gaussian noise with $\sigma_{noise} = 0.001$ in both the log of the prior scale $\log \sigma_p$ and the posterior scale $\log \sigma_q$ helps network's generalization.

**Results.** Table 3 shows the performance of the proposed model both among other VAEs and other purely generative models. As we can see, attention-guided variational inference is conducive to building more informative latent spaces, rendering the model the best performing in its class. This claim is further corroborated by qualitative results in Appendix E. We can see there that the attention-driven skip connections mitigate posterior collapse by activating early layers in the hierarchy. Moreover, our model closes the performance gap between VAEs and expensive generative models that rely on fully-autoregressive distributions. Finally, it is possible our results in Table 3 could be improved by considering deeper hierarchies.

Table 3: **CIFAR-10** (Krizhevsky et al., 2009) **performance on the test set.** The marginal log-likelihood is estimated with 100 importance samples. A shallower Attentive VAE outperforms all state-of-the-art VAEs with or without autoregressive components. Attentive VAE performs on par with fully autoregressive generative models. However, it permits fast sampling that requires a single network evaluation per sample as opposed to $D$, where $D$ the dimension of the data distribution.

| Model | VAE | Depth ($L$) | Autoregressive Decoder | $-\log p(\boldsymbol{x}) \leq$ (bits/dim) $\downarrow$ |
|---|---|---|---|---|
| **Attentive VAE (ours) trained for 400 epochs** | ✓ | 16 | ✗ | **2.82** |
| **Attentive VAE (ours) trained for 500 epochs** | ✓ | 16 | ✗ | **2.81** |
| **Attentive VAE (ours) trained for 900 epochs** | ✓ | 16 | ✗ | **2.79** |
| Very Deep VAE (Child, 2020) | ✓ | 45 | ✗ | 2.87 |
| NVAE (Vahdat & Kautz, 2020) | ✓ | 30 | ✗ | 2.91 |
| BIVA (Maaløe et al., 2019) | ✓ | 15 | ✗ | 3.08 |
| IAF-VAE (Kingma et al., 2016) | ✓ | 12 | ✗ | 3.11 |
| $\delta$-VAE (Razavi et al., 2019a) | ✓ | | ✓ | 2.83 |
| PixelVAE++ (Sadeghi et al., 2019) | ✓ | | ✓ | 2.90 |
| Lossy VAE (Chen et al., 2017) | ✓ | | ✓ | 2.95 |
| MAE (Ma et al., 2019) | ✓ | | ✓ | 2.95 |
| PixelCNN++ (Salimans et al., 2017) | ✗ | | ✓ | 2.92 |
| PixelSNAIL (Chen et al., 2018) | ✗ | | ✓ | 2.85 |
| Image Transformer (Parmar et al., 2018) | ✗ | | ✓ | 2.90 |
| **Sparse Transformer** (Child et al., 2019) | ✗ | | ✓ | **2.80** |

In Table 4, we compare the computing demands for training our model with the one of the other two most competitive VAEs presented in Table 3. As we see, our 16-layer VAE requires almost $3/5$ of the GPU hours needed for training a 30-layer NVAE, while still outperforming the more efficient VAE of Child (2020). In this work, we implement full rank matrices for the self-attention scores of the spatially non-local residual cells. Efficient attention approximations such as those proposed by Child et al. (2019) and Choromanski et al. (2021) could further reduce the computational requirements of our models and are left as future work. This suggestion is also bolstered by our findings in Appendix F, where we observe that the spatial attention maps are sparse and highly structured.

Table 4: **Comparison of the computational requirements for training deep state-of-the-art VAE models.** All models are trained on 32GB V100 GPUs. The additional cost for computing the attention scores is compensated by the smaller number of stochastic layers in the hierarchy without sacrificing the generative capacity of the model, see Table 3.

| Model | batch size / GPU | # GPUs | Training Time | Total GPU hours |
|---|---|---|---|---|
| Attentive VAE (ours), 400 epochs | 32 | 4 | 68 hours | 272 |
| Attentive VAE (ours), 500 epochs | 32 | 4 | 84 hours | 336 |
| Attentive VAE (ours), 900 epochs | 32 | 4 | 152 hours | 608 |
| NVAE | 32 | 8 | 55 hours | 440 |
| Very Deep VAE | 32 | 2 | 6 days | 288 |

## 3.2 ABLATION STUDY

To verify the effectiveness on the performance of the different attention schemes proposed in Section 2, we conduct comprehensive ablation studies on architectures of a variable number of stochastic layers on the CIFAR-10 dataset. We incrementally add different combinations of the proposed attention components of Sections 2.4, 2.5, 2.6 in the NVAE architecture and we evaluate their impact on the estimated marginal likelihood reported in bits per dimension. Our results are shown in Table 5. We make the following observations:

- Regardless of the number of stochastic layers in the hierarchy, we observe improvement over the baseline model (Case 1 vs Case 8 and Case 9 vs Case 16 in Table 5).

- We notice that a 8-layer architecture that employs an attention mechanism in both the generative network and the inference counterpart of the model (Case 8 in Table 5) almost reaches the performance level of the baseline 16-layer architecture (Case 9 in Table 5). This fact indicates that our model results in a better utilization of the latent space. This is orthogonal to other works which aim at improvement by stabilizing deeper architectures instead.

- The effect of the spatially attentive stochastic layers is larger in the shallower architecture (Case 1 vs Case 2 and Case 9 vs Case 10 in Table 5). This may be because some loss of the latent information is compensated by additional layers. However, the total improvement yielded by the depth-wise attention modules is larger in the deeper architecture, highlighting the importance of the inter-layer connectivity when constructing the prior and the posterior of each layer. The inter-layer connectivity is also visualized in Figures 10, 11 in Appendix F, where it is confirmed that the model learns to attend to context information that lies in distant layers of the model to improve performance.

Finally, supplementary ablations on other architectural choices of our model are provided in Appendix D.

Table 5: **A comparison of the effect of the attention operations in a deep variational model on the CIFAR-10 dataset.** The NVAE (Vahdat & Kautz, 2020) is used as a baseline case (no attention operation).

| Depth ($L$) | Case | Depth-wise generative Section 2.4 | Depth-wise inference Section 2.5 | Non-local residual cells Section 2.6 | # Parameters ($\times 10^6$) | $-\log p(\boldsymbol{x}) \leq$ (bits/dim) $\downarrow$ |
|---|---|---|---|---|---|---|
| 8 | 1. | ✗ | ✗ | ✗ | 39.473 | 3.062 |
| | 2. | ✗ | ✗ | ✓ | 44.585 | 2.998 |
| | 3. | ✗ | ✓ | ✗ | 44.716 | 3.019 |
| | 4. | ✗ | ✓ | ✓ | 50.471 | 2.98 |
| | 5. | ✓ | ✗ | ✗ | 46.3 | 3.034 |
| | 6. | ✓ | ✗ | ✓ | 52.151 | 2.985 |
| | 7. | ✓ | ✓ | ✗ | 51.925 | 2.986 |
| | 8. | ✓ | ✓ | ✓ | 58.451 | 2.963 |
| 16 | 9. | ✗ | ✗ | ✗ | 79.206 | 2.939 |
| | 10. | ✗ | ✗ | ✓ | 89.602 | 2.912 |
| | 11. | ✗ | ✓ | ✗ | 90.916 | 2.918 |
| | 12. | ✗ | ✓ | ✓ | 102.744 | 2.872 |
| | 13. | ✓ | ✗ | ✗ | 92.983 | 2.92 |
| | 14. | ✓ | ✗ | ✓ | 104.882 | 2.905 |
| | 15. | ✓ | ✓ | ✗ | 105.541 | 2.861 |
| | 16. | ✓ | ✓ | ✓ | 118.966 | 2.823 |

## 4 RELATED WORK AND DISCUSSION

Our work builds on recent deep variational autoencoders (Vahdat & Kautz, 2020; Child, 2020), which first perform a fully deterministic bottom-up pass followed by a top-down pass involving both stochastic and deterministic features, as first proposed by Kingma et al. (2016). This hybrid top-down pass enables fully connected factorizations of the joint prior and posterior distributions, according to which the latent variables of each layer are conditioned on the latent variables of all the layers above in the hierarchy. However, modeling such conditional dependencies relies on local architectures which assume connectivity only between successive layers. We improve on prior work by introducing attention-guided mechanisms to actively discover long-range statistical dependencies in the latent space. We build on top of NVAE (Vahdat & Kautz, 2020) to demonstrate our method, but our suggestions are generic and could be applied to deeper and thinner hierarchies such as that of Child (2020), or the BIVA hierarchy which extends the deterministic bottom-up pass to latent variables (Maaløe et al., 2019). These configurations could further improve the results reported in Tables 2, 3. Moreover, the proposed model is amenable to other tools for improved variational inference such as the use of an autoregressive posterior (or prior) distribution (Kingma et al., 2016), a second-stage ancenstral sampling by an auxiliary VAE that learns the aggregated posterior (Dai & Wipf, 2019), or regularization techniques that promote congruence between latent representations of the original and transformed inputs (Sinha & Dieng, 2021).

Vector Quantized Variational Autoencoders (VQ-VAEs) (Oord et al., 2017; Razavi et al., 2019b) learn discrete latent representations. However, the training objective of these models does not formally constitute a lower bound on the marginal log-likelihood. Although VQ-VAEs have exhibited high-quality generative capability on high-resolution images, they base their performance on fully autoregressive PixelCNN priors (Van Den Oord et al., 2016) which render them too slow to sample from in case of large latent spaces. Most related to ours is the work of Esser et al. (2021). The authors use a fully autoregressive transformer in the latent space. However, they manage to handle low-dimensional latent spaces, leading to log-likelihoods infererior to that of Razavi et al. (2019b). Scaling this model to larger latent spaces, by leveraging the decomposed, attention-guided latent hierarchy proposed in this work could be an interesting future research direction.

Several works on generative modeling leverage self-attention mechanisms (Parmar et al., 2018; Chen et al., 2018; 2020; Child et al., 2019). However, these models are autoregressive, yielding conditional per-pixel distributions, and apply attention in the pixel space. Therefore, as already discussed, they are slow to sample from. Our work has log-likelihood performance comparable to that of these works, while using a non-autoregressive data distribution. The work of Tulsiani & Gupta (2021) uses a transformer-based encoder to learn pixel distributions conditioned on a small set of observed pixel values at arbitrary locations. Diffusion models (Ho et al., 2019; Kingma et al., 2021) can be viewed as VAEs and achieve impressive generative quality. However, similarly to the autoregressive models, sampling from them requires multiple network evaluations. Morever, their performance when trained with data augmentation (Kingma et al., 2021) can be similar to that of deep regularized VAEs (Sinha & Dieng, 2021). Finally, the work by Zhang et al. (2019) boosts GANs by incorporating non-local blocks (Wang et al., 2018) in the generator and the discriminator.

Future research includes investigating the effect of the proposed depth-wise attention mechanism on ResNet architectures for different tasks and efficient attention approximations specific to deep variational inference.

## 5 CONCLUSION

We have presented deep attentive VAE, the first attention-driven framework for variational inference in deep probabilistic models. We argue that the expressivity of current deep probabilistic models can be improved by selectively prioritizing statistical dependencies between latent variables that are potentially distant from each other. We introduced a scalable factorized attention scheme that first discovers intra-layer correlations, succeeded by inter-layer attention operations. Moreover, the generative model is allowed to query its posterior counterpart based on the available latent information to identify the data-dependent representations most informative for inference. We extensively evaluated the proposed architecture on multiple public datasets and showed that our model outperforms all existing deep VAEs, while requiring significantly less training time.

To aid reproducibility of the results and methods presented in our paper, we made source code to reproduce the main results of the paper publicly available, including detailed instructions; see our github page: `https://github.com/ifiaposto/Deep_Attentive_VI`.

## 6 ACKNOWLEDGEMENTS

We thank anonymous reviewers for their careful reading of our manuscript and their thoughtful comments and suggestions that helped us improve the quality of the paper. We also thank Christos Louizos for helpful pointers to prior works on VAEs, Katerina Fragkiadaki for helpful discussions on generative models and attention mechanisms for computer vision tasks, Andrej Risteski for insightful conversations on approximate inference, and Jeremy Cohen for his remarks on a late draft of this work. We are also very grateful to Radium Cloud for granting us access to computing infrastructure that enabled us to scale up our experiments. This material is based upon work supported by the Defense Advanced Research Projects Agency under award number FA8750-17-2-0130, and by the National Science Foundation under grant number 2038612. The first author acknowledges support from the Alexander Onassis Foundation and from A. G. Leventis Foundation. The second author is supported by the National Science Foundation Graduate Research Fellowship Program under Grant No. DGE1745016 and DGE2140739. Any opinions, findings, and conclusions or recommendations expressed in this material are those of the author(s) and do not necessarily reflect the views of the National Science Foundation.

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
