# OpenReview forum: "Deep Attentive Variational Inference"
_ICLR.cc/2022/Conference — ICLR 2022 Poster_

### Official Review · Reviewer_VA9m · 2021-10-30

**Correctness:** 3
**Technical Novelty And Significance:** 3
**Empirical Novelty And Significance:** 3
**Recommendation:** 6
**Confidence:** 3

**Main Review:**

Pros:
1. This was the first work where the attention mechanism was proposed to carry out the deep hierarchical VAE. By explicitly modeling the local and global interactions in latent space, an expressive variational distribution was constructed with probabilistic justification.
2. This paper provided the experimental evidence to show that the previous method on expressive variational model via designing the deep hierarchy of interdependent latent variables would incur the problem of diminishing returns.
3. The experimental justification with the comparison over different types of VAE models was sufficient.
4. Computational cost was reduced when compared with the previous SOTA model based on NVAE.

Cons:
1. It is suggested to provide an algorithm to enhance the comprehension of the detailed training procedure of the proposed method.
2. The experiments did not show the generated samples the authors obtained. The quality of the generated images could not be evaluated.
3. Ablation study was only conducted on the effect of non-local information. The ablation studies different normalization and activation functions are required.

**Summary Of The Paper:**

This paper presented a variational inference with attention mechanism. A deep latent variable model was proposed.

**Summary Of The Review:**

An interesting work was proposed with experimental justification. The experiments can be furthered strengthened.

---

> ### Author Response · Authors · 2021-11-19
> **Thank you for your positive review and constructive criticism!**
>
> We appreciate that you found the work interesting, well motivated, and experimentally convincing. We will respond to each of your points directly in the comments below.  We hope that our additions will convince you to raise your score. If not, please do let us know by replying to this post. We would be happy to clarify things further.
>
> *"It is suggested to provide an algorithm to enhance the comprehension of the detailed training procedure of the proposed method.''*
>
> This is a great suggestion. We also believe that an algorithm summarizing the training procedure could clarify the aspects of the proposed model as well as subtleties in the inference procedure. Therefore, in **Appendix A.3 in Algorithms 1, 2, 3, we provide the steps involved in computing the NELBO loss of Equation (2)**. The algorithms are accompanied by explanatory text. Please do let us know if you feel there are points in the model and/or inference that are still unclear. Moreover, upon acceptance of the paper we will release the source code reproducing all experiments in the paper to ensure that it will be broadly utilized by the research community.
>
> *"The experiments did not show the generated samples the authors obtained. The quality of the generated images could not be evaluated."*
>
> We have **added** the following **qualitative assessments of the proposed model** in the Appendix:
>
> 1) In **Appendix [D], in Figures 4, 5**, we plot the $\boldsymbol{log D_{KL}}$ **per layer**. These Figures experimentally show that the attention driven couplings help keep the latent variables of all layers active significantly mitigating posterior collapse compared to current deep VAE architectures which still exhibit unbalanced $log D_{KL}$ between the top and the lower layers.
>
> 2) In **Appendix [E], in Figures [6]-[10]**, we visualize the **attention maps of the model on some test images**. These plots corroborate our initial hypothesis that informative latent context information may lie far beyond in the hierarchy (large attention scores for entities distant from each other) either layer-wise (Figures 6-7) or in the spatial domain at the same layer (Figures 8-10).
>
> 3) In **Appendix F, in Figures 11, 12**, we plot **reconstructions of sample test images as well as novel samples** generated by our model.
>
> *"Ablation study was only conducted on the effect of non-local information. The ablation studies different normalization and activation functions are required.''*
>
> This is also a great suggestion. We apologize for the omission in the initial version of our manuscript. We have added a variety of experiments to explore the impact of several architectural choices on the performance in **Appendix C**. In particular,
>
> 1) In **Table [7]**, we study the **effect of weight normalization**.
>
> 2) In **Table [8]**, we study the **effect of batch normalization**.
>
> 3) In **Table [9]**, we study the **effect of the non-linearity** applied in the ResNet cells.
>
> 4) In **Table [10]**, we study the **effect of the depth-wise normalization** scheme proposed in Equations (6), (7) in the main paper.
>
> We should also note that while varying any of the aforementioned hyperparameters, we did not observe any instability issues when training our model. Therefore, the **strongly connected couplings among the layers help stabilize training of our model** and **make it robust to the choice of these hyperparameters**.

---

### Official Review · Reviewer_NQd9 · 2021-11-02

**Correctness:** 3
**Technical Novelty And Significance:** 3
**Empirical Novelty And Significance:** 4
**Recommendation:** 8
**Confidence:** 4

**Main Review:**

## 1. Strengths
- a. The introduction of the layer-wise attention mechanism for deep VAEs is novel, and its effectiveness is supported by empirical results
- b. The introduction of non-local attention within layers is effective, and supported by experiments
- c. The overall architecture (incl. normalisation and scaling tricks) challenges the current sota deep VAEs using fewer layers and shorter training.
- d. Literature in deep VAEs seems to be well known by the authors and sufficiently cited

## 2. Weaknesses
- a.  The paper lacks structure and clarity
- b. The paper lacks a more qualitative study of the model:
    - it would be interesting to see what layers the layer-wise attention mechanism attends to.
    - it would be great to understand how this model uses the latent variables, for instance by measuring the KL divergence at each layer, as done in the previous work (LVAE, BIVA) (connection to "posterior collapse").
- c. Experiments are limited to CIFAR-10, larger scaler experiments (i.e. ImageNet) would be beneficial to the paper. It is not guaranteed that such an architecture would translate in the same gains for larger datasets (i.e. ImageNet).

## 3. Clarification needed:
- a. Table 5: I interpreted the column "non-local layers" as using "attention across layers", I hope I was right. The nomenclature needs to be improved.
- b. Is the layer-wise attention mechanism specific to deep VAEs, or can it be more generally applied to ResNet architectures?
- c. Section 2.3 (paragraph cited bellow): I get the idea, but unless demonstrated, this remains a hypothesis.
    >"... in practice the network may no longer respect the factorization of the prior $p(z)=\prod_{l} p(z_{l} \mid z_{<l})$ leading to diminished performance gains as shown in Table 1":

## 4. Minor comments / suggestions
- a. The main contributions are introducing two types of attention for deep VAEs, it might help to describe them in a separate section, and only then describe the generative and inference models. Right now the description of the layer-wise attention mechanism is scattered across sections 2.3 and 2.4.
- b. tricks like normalisation or feature scaling could be referenced in a separate section.
- c. eq8: you might want to cite ReZero [1] here
- d. Fig 1. a: the lack of arrows going from the activations $(h_l, k_l^q)$ to the attention block $(\mathcal{A}(...))$ was confusing on the first read
- e. It would be better practice to report likelihoods for multiple random seeds
- f. Typo in section 2.1: "both $q(z|x)$ and $p(x)$ are fully factorized gaussian..." -> "both $q(z|x)$ and $p(z)$ are fully factorized gaussian..."

[1] Bachlechner, T., Prasad Majumder, B., Mao, H. H., Cottrell, G. W., and McAuley, J., “ReZero is All You Need: Fast Convergence at Large Depth”, <i>arXiv e-prints</i>, 2020.


**Summary Of The Paper:**

This paper improves the architecture of deep VAEs using the attention mechanism.

The attention mechanism is used in two ways:
1. layer-wise attention (attending stochastic feature maps which are conditioned on other variables within the hierarchy, interpreted as a mixture of skip connections)
2. non-local attention (attention across the spatial dimensions, increases the size of receptive field)

The authors demonstrate the effectiveness of their architectural changes by challenging current sota deep VAEs on MNIST, OMNIGLOT and CIFAR-10. Notably, they outperform the state-of-the-art methods (in likelihood) on CIFAR-10 using fewer layers and fewer GPU hours.

The authors provide an extensive ablation study, showing the impact of each type of attention on the training performances (test likelihood on CIFAR-10).

**Summary Of The Review:**

I have enjoyed reading your paper, nice work! The introduction of the layer-wise attention mechanism is a great contribution. Using non-local blocks within the model is less novel, but this is still new for such a model.

The results on MNIST, Omniglot and CIFAR-10 demonstrate the effectiveness of the method, and the ablation study clearly shows the positive effect of the two attention modules (layer-wise and spatial).

Experiments remain however limited. Large-scale experiments (ImageNet) and qualitative studies (inspective learned attention) would be a great addition. They would give the reader a better understanding of the impact of the attention modules. The paper overall writing quality and structure need to be improved.

---- edit 26/11
Raised my score to 8.

---

> ### Author Response · Authors · 2021-11-19
> **Thank you for recognizing the significance of our contributions and your insightful suggestions! (2/2)**
>
> *Regarding your note on the structure and clarity of the paper:*
>
> We agree that restructuring the paper with separate sections for each of our technical contributions would help highlight the novelties and improve comprehensibility of our work. This is a great suggestion and it will be incorporated in the updated version of the manuscript. To further improve clarity, in **Appendix A.3 we have provided algorithms describing in detail all parts of our model**.
>
> *Regarding your clarification requests:*
>
> *"Table 5: I interpreted the column "non-local layers'' as using "attention across layers'', I hope I was right.''*
>
> Thank you for identifying this point of confusion! This interpretation is not correct, so we will be sure to reword in the updated version for clarity. Here, **"Non-local layers'' pertains to the spatially non-local layers (section 2.5)**. Similarly, "Non-local generative'' refers to section 2.3 and "Non-local inference'' refers to section 2.4.
>
> *''Is the layer-wise attention mechanism specific to deep VAEs, or can it be more generally applied to ResNet architectures?'''*
>
> This is a great direction for future research. We believe that the depth-wise attention scheme proposed in this work could be also applied to ResNet architectures for different tasks. **Attention-guided versions of densely connected convolutional networks [5] (Attentive DenseNets) is left as a very promising future work**.
>
> *"I get the idea, but unless demonstrated, this remains a hypothesis: `... in practice the network may no longer respect the factorization of the prior leading to diminished performance gains as shown in Table 1''*
>
> **This is empirically reinforced by Figure 6**. Note that the context passed to layer $l$, $\boldsymbol{c}^p_{l-1}$, already contains by construction latent information from $\boldsymbol{z}_{<l}$.
>
> The fact that the model selects to activate couplings with layers that lie far above in the hierarchy indicates that useful latent information is attenuated in $\boldsymbol{c}^p_{l-1}$. If this was not the case, the corresponding connection would be dormant by assigning a small attention score.
>
>
> *"c. eq8: you might want to cite ReZero [1] here''*
>
> Thank you for the reference. We will cite accordingly in the updated version of the manuscript.
>
> *"Fig 1. a: the lack of arrows going from the activations to the attention block was confusing on the first read''*
>
> We have **improved Figure 1b**, where we now denote the order of the computations in the modules of each layer. Incoming arrows annotate inputs, and outgoing arrows indicate outputs. The inputs of each module are either outputs from a preceding module in the same layer or inputs coming from the output of a previous layer. These entities are matched by name.
>
> *"It would be better practice to report likelihoods for multiple random seeds''*
>
> This is a fair point. As we mention above, our experimental resources are limited---for the original submission, we dedicated our resources to expanding our experiments to multiple datasets and to performing ablation studies on the architecture. We will address this in the camera-ready version of the paper. Please note that **during the fine-tuning of our model, we did not notice large variance on the NELBO reached**.
>
> Finally, thanks for the typos (and points of confusion) that you've pointed out! We have fixed these in the updated manuscript.
>
> [1] Asperti, Andrea, Davide Evangelista, and Elena Loli Piccolomini. "A Survey on Variational Autoencoders from a Green AI Perspective." SN Computer Science 2.4 (2021): 1-23.
>
>   [2]Schwartz, R.; Dodge, J.; Smith, N.A.; Etzioni, O. Green ai. Commun. ACM 2020.
>
>   [3] Rewon Child. Very deep vaes generalize autoregressive models and can outperform them on images.
>
>   [4] Arash Vahdat and Jan Kautz. Nvae: A deep hierarchical variational autoencoder.
>
>   [5] Huang, G., Liu, Z., Van Der Maaten, L. and Weinberger, K.Q., 2017. Densely connected convolutional networks. In Proceedings of the IEEE conference on computer vision and pattern recognition (pp. 4700-4708)

---

> > ### Comment · Reviewer_NQd9 · 2021-11-26
> > **Great improvements**
> >
> > Thank you for the detailed reply. The paper has greatly benefitted from the reviewers' feedback, which the authors have successfully used to improve the paper. I am confident that this paper introduce significant contributions to the field. The contributions are supported by a robust set of experiments. I raise my score to 8.
> >
> > > Regarding larger-scale experiments
> >
> > I understand and agree with the cost of running ImageNet experiment being prohibitively expensive. The experiments on CelebA are a good solution, however, the results must be added to the camera-ready version.
> >
> > > Regarding the additional qualitative studies
> >
> > Great visualisation and valuable addition to the paper, that answers my concerns. The visualisation of the latent activations (fig 4) clearly shows that using attention enables activating all layers. The visualisation of the attention maps (fig 6) shows that attention strategies are learned.
> >
> > > Improving the structure and clarity
> >
> > Changes are satisfactory
> >
> > > likelihood for multiple seeds
> >
> > Great, deep VAEs usually don't suffer from high variance, but it would be great to add an estimate of the variance to the camera-ready.

---

> ### Author Response · Authors · 2021-11-19
> **Thank you for recognizing the significance of our contributions and your insightful suggestions! (1/2)**
>
>  Your detailed feedback helps us to improve the quality of our work. Our comments addressing all of your concerns can be found below. We hope that our additions and clarifications will convince you to raise your score. If you still have hesitations, please let us know.
>
>   *Regarding larger scale experiments:*
>
>   We have conducted a larger-scale experiment on the **CelebA dataset** (64x64x3 images) by using only the depth-wise attention proposed in the paper. We use 3 scales of latent variables with an adaptive number of layers (as in the NVAE) per scale (12, 6, 4 per scale in total **22 layers compared to the 35 layers of NVAE**). We have managed to reach **2.07 bits/dimension** (without normalizing flows) by training on 4 GPUs for 3.5 days. This is on par with the NVAE performance (2.04 bits/dimension) which is accomplished by training for 92 hours on 8 GPUs. The above mentioned performance is no small feat from a GreenAI perspective [1], [2].
>
>   The performance can be further improved by fine-tuning, longer training, and use of spatial attention as proposed in our work. Once the architecture is finalized, we would like to add it in the camera-ready of the version of the paper. Given that **our current experiments pertain to datasets of increasing difficulty, we have strong evidence that the effect of the attention is larger on deeper hierarchies and more complex datasets**.
>
>   Regarding scaling to Imagenet, please note that works on deep VAEs (e.g. [3], [4]) which report results on such a large and computationally-demanding benchmark make use of a very large number of high capacity GPUs (the above references use a total of 32 and 24 GPUs with 32 GB of memory, respectively). These computing resources are not easy to acquire (according to current AWS pricing access to an 8-GPU server with 32GB GPUs would cost 32\$/hour), rendering these experiments not reproducible by a large part of the research community.
>
>   **While we agree that our submission would be further strengthened by reporting results on ImageNet, such resources are simply not available to us currently; we do however feel that our extensive experiments on more moderately-sized datasets such as CIFAR and CelebA present sufficient evidence for the effectiveness of our architecture.**
>
> *Regarding additional qualitative studies:*
>
> We have added the following qualitative experiments in **Appendix D, E** which confirm the hypothesis made in the main paper.
>
> 1. In **Figures [4], [5]** in **Appendix D**, we plot the $\boldsymbol{log D_{KL}}$ **per layer** during training for a deep VAE without attention (Figure 4) and with attention (Figure 5). We show that the long-range couplings evenly distribute the $log D_{KL}$ between all layers keeping them active (and therefore better utilized). This is not the case for a deep VAE without attenion, in which case the number of active latent variables is significantly larger for the layers which are lower in the hierarchy. In other words, **the proposed model mitigates effectively posterior collapse**.
>
> 2. In **Figures [6], [7], [8], [9], [10]** in **Appendix E**, we provide the **attention patterns** (depth-wise attention in the generative network, depth-wise attention in the inference network, spatial attention in selected layers). These plots confirm that **the model learns to attend to context information that lies in distant parts of the model to improve performance**. In case the context information $\boldsymbol{c}^p_{l-1}$ would suffice for inference when building the prior and posterior distribution of layer $l$, the skip connections would not have been activated by layer $l$ yielding small attention scores. These plots provide also further insights. In Figures [8], [9], [10], we can see that the **spatial attention patterns are highly structured and sparse**. This fact suggests a direction for designing sparse attention models tailored to variational inference and deep probabilistic models. We welcome further research on this area.

---

### Official Review · Reviewer_DYsA · 2021-11-02

**Correctness:** 3
**Technical Novelty And Significance:** 3
**Empirical Novelty And Significance:** 3
**Recommendation:** 8
**Confidence:** 3

**Main Review:**

**Relevance**
Deep VAEs are an active area of research and progress is commonly quantified by the marginal likelihood on standard benchmark datasets. Hence improvements in that regard make this a relevant contribution.

**Novelty**
Multi-layer structure as well as attention are popular techniques in their respective fields, but their combination is novel as far as I am aware. The specific method in this paper is a non-trivial combination of these, so I would consider this a solid, but not outstanding paper in terms of novelty.

**Clarity**
Overall clear. A fair bit of notation, but that seems unavoidable. There are a couple of figures to visualize the computational graph structure, which is helpful. Perhaps an algorithm box for inference and generative paths as well as the loss calculation would be useful as well.

Some terminology is used a bit loosely in my view, in particular “local” and “global”. For me as a reader with more of a general probabilistic ML background, these refer to per-datapoint and shared variables across all data respectively (so in VAEs the latent variables are local, if we did inference over the weights those would be global). But this might be standard in the more closely related literature.

**Empirical evaluation**
The benchmarks (MNIST, Omniglot, CIFAR10) and metrics (marginal likelihood) are standard as far as I’m aware. There is an extensive ablation study, although it is not obvious to me why the combination of a non-local generative and inference model without non-local layers is not considered. The paper suggests that the proposed method could be combined with autoregressive generative models, adding such an experiment would strengthen the empirical contribution of the paper.

**Other notes and questions**
* First sentence final paragraph of 2.1: p(x) should be p(z)?
* Is it possible to further increase the depth of the architecture when using attention? It would be useful to have an equivalent table to Tab 1 to support the claim that the attention-based scheme does not suffer from diminishing returns on increasing depth to the same degree as the baseline. If there are limitations due to the quadratic complexity in depth on current hardware, this should be mentioned explicitly (although I might have missed it).
* The formatting of the references is extremely inconsistent, it seems like the bibtex entries were copy-pasted from google scholar without change. Please be consistent in venue names, title capitalization, abbreviations of middle names etc.

**Summary Of The Paper:**

This paper proposes a novel attention-based architecture for deep VAEs that facilitates dependencies between non-neighbouring layers and reports improvements on the marginal likelihood over recent related methods on MNIST, Omniglot and CIFAR10.

**Summary Of The Review:**

I’m not deeply familiar with the recent literature on (more sophisticated) VAE architectures, but the method, while a combination of existing techniques and ideas, appears to be novel and I would expect the empirical results to be of interest to the community. Therefore I am **leaning towards accept**.

**Post-rebuttal note**: Increased score to accept.

---

> ### Author Response · Authors · 2021-11-19
> **Thank you for your encouraging feedback and your thorough corrections! (2/2)**
>
> *"I’m not deeply familiar with the recent literature on (more sophisticated) VAE architectures...interest to the community.''*
>
>   Deep VAEs are notoriously challenging to stabilize and train. There are some recent, very impactful works (for example, see [1], [2]) that focus exclusively on training very deep VAEs by using regularization techniques such as gradient clipping or spectral regularization and combining effectively various implementation aspects (type of cells in the encoder/decoder, momentum of batch normalization, activation functions to be used). Specific to our model, we would like to highlight some implementation aspects without which we would not have managed to scale the model and achieve SOTA performance. In particular,
>
>   * **Naive implementation of attention,** where the keys and the queries are not direct outputs of the ResNet modules produced along with the context vector $\boldsymbol{c}^p_l$, but computed at a second stage, by a linear transformation of $\boldsymbol{c}^p_l$ implemented as 1x1 convolution, **would render the model too 'bulky' to train**.
>
>   * Similarly, the normalization scheme proposed in Equations 6, 7 is critical for scaling up the models. **Current attention normalization schemes** involve additional convolutions [3], which also **render deep VAEs difficult to train and had to be adapted**.
>
>   Therefore, we feel that **the architecture itself has its own merit**. We would also like to underscore the following **technical contributions of our work**:
>
> 1) The formulation of the **depth-wise attention mechanism, and its normalized version**, enables a very deep receptive field for a layer in the hierarchy. Naive 3D attention mechanisms to account for the 'third' dimension in the latent space, which is a byproduct of the factorization of the distributions and its organization in layers, would not be applicable in such architectures. As also suggested by Reviewer 3, it would be interesting to apply the proposed mechanism in ResNet architectures on different tasks, yielding attention driven dense connections (DenseNet) between the layers.
>
> 2) **Additional coordination between the prior and the posterior occurs when forming the conditioning factors of the posterior** (the generative network actively discovers the data features which complementary to the latent features yield the most descriptive context for the posterior). Novel works such as LadderVAE and NVAE keep the prior and the posterior in balance via their parametrization only (with a precision-weighted combination and with a residual correction  respectively).
>
>   We are planning to restructure the paper in the revised version to highlight these contributions further. Finally, **we have further strengthened the empirical novelty and significance of our model**:
>
>   1) After training the model for more epochs (with learning rate warm restarts), we have pushed the performance further to 2.79 bits per dimension surpassing the sparse transformer (see **updated version of Table 3**) and **completely closing the gap with expensive autoregressive models**.
>
>   2) **In Appendix D, Figure 5**, we show empirically that **our model tackles effectively posterior collapse** -a long lasting deficiency of deep VAEs- as also shown in Figure 4.
>
>  Regarding the typographical issues you've pointed out, we want to thank you for bringing these items to our attention! You are correct regarding the typo in paragraph 2.1. We will also reformat the bibliography for consistency in the updated version of the manuscript.
>
>
>   [1] Rewon Child. Very deep vaes generalize autoregressive models and can outperform them on images.
>
>   [2] Arash Vahdat and Jan Kautz. Nvae: A deep hierarchical variational autoencoder.
>
>   [3] Chen M, Radford A, Child R, Wu J, Jun H, Luan D, Sutskever I. Generative pretraining from pixels.

---

> > ### Comment · Reviewer_DYsA · 2021-11-24
> > **Response**
> >
> > Thank you for the extensive updates, which nicely round off the paper, and elaborating on the technical novelty of the architecture. As none of the other reviewers has raised any critical issues, I am happy to increase my score.

---

> ### Author Response · Authors · 2021-11-19
> **Thank you for your encouraging feedback and your thorough corrections! (1/2)**
>
> Thank you very much for your thorough review. We are glad to hear that you found the paper relevant to the scope of the conference, solid, clear and of practical utility to the research community. We hope the following points address your concerns. If not, please do let us know by replying to this post. We would be happy to clarify things further.
>
> *"Perhaps an algorithm box for inference and generative paths as well as the loss calculation would be useful as well.''*
>
> This is a great suggestion. We also believe that an algorithm summarizing the training procedure could clarify the aspects of the proposed model as well as subtleties in the inference procedure. Therefore, in **Appendix A.3 in Algorithms 1, 2, 3, we provide the steps involved in computing the NELBO loss of Equation (2)**. The algorithms are accompanied by explanatory text. For the generative part, the bottom-up pass and the formation of the posterior in Algorithm 3 is skipped while the prior is sampled instead. Please do let us know if you feel there are points in the model and/or inference that are still unclear. Moreover, upon acceptance of the paper we are planning to release the source code reproducing all experiments in the paper to ensure that it will be broadly utilized by the research community.
>
> *"Some terminology is used a bit loosely in my view, in particular “local” and “global”... this might be standard in the more closely related literature.''*
>
> This is a very helpful suggestion. We will reword for clarity in the revised version of the manuscript.
>
> *"There is an extensive ablation study, although it is not obvious to me why the combination of a non-local generative and inference model without non-local layers is not considered.''*
>
> We apologize for the omission. We have **updated Table 5 to include this case**.
>
> *"Is it possible to further increase the depth of the architecture when using attention?...this should be mentioned explicitly.''*
>
> This is an excellent suggestion! In **Appendix D, Table 11**, we conducted an analysis similar to that of Table 1, up to 16 layers. We show that **the suggested model eliminates diminishing returns present in current deep VAEs**. Unfortunately, due to computational limitations (we have access only up to 32GB GPUs) we cannot train larger models. The quadratic complexity (with respect to H, W) is already discussed in Section 2.5. However, we note that this pertains only to the spatial attention. Moreover, the model exhibits highly-structured attention maps, as visualized in Figures 8-10 in Appendix E. Therefore, it is amenable to sparse attention approximations which could push the performance of the model further. We do welcome research towards this direction.

---

### Official Review · Reviewer_5Yd5 · 2021-11-02

**Correctness:** 4
**Technical Novelty And Significance:** 2
**Empirical Novelty And Significance:** 3
**Recommendation:** 8
**Confidence:** 3

**Main Review:**

Strength:
1. So far, this is the first work that I know of that successfully combined transformer type of attention to VAE models. Even though it is a combination of two existing models, I believe it still provides benefit and value to the VI research community.
2. Evaluation showed strong superior performance compared with a wide range of baseline models.
3. An ablation study is included to justify each components of the proposed model.
4. The paper is well written and the proposed model is easy to understand.

Weakness:
This is a solid paper, no major weakness. The only concern I have is regarding the novelty of the proposed framework. However, as discussed before, I believe this still adds value to the variational inference research community.

**Summary Of The Paper:**

This paper identifies a common problem in previous VAE related models: adding more stochastic layers to an already very deep model yields small predictive improvement while substantially increasing the inference and training time. Therefore, a new model that proposes to use attention mechanisms to build more expressive variational distributions in deep probabilistic models by explicitly modelling both local and global interactions in the latent space is proposed. The model is evaluated on standard dataset MNIST and OMNIGLOT, and showed superior performance against a wide range of baseline models.

**Summary Of The Review:**

Overall, this is a pretty solid paper. Please refer to the previous section for detailed discussion.

---

> ### Author Response · Authors · 2021-11-19
> **Thank you for providing positive comments on our manuscript and for your accurate summary!**
>
> We are encouraged by the fact that you think our work is of utility and value to the community.
>
>  In order to **broaden the empirical impact of our paper**, and in response to your comments, we have extensively supplemented the experiments:
>
>   1) After training the model for more epochs (with learning rate warm restarts), we have pushed the performance further to 2.79 bits per dimension surpassing the sparse transformer (see **updated version of Table 3**) and **completely closing the gap with expensive autoregressive models**.
>
>   2) In **Appendix D, Figure 5**, we show empirically that **our model tackles effectively posterior collapse** -a long lasting deficiency of deep VAEs- as also shown in Figure 4.
>
>  Please also keep in mind that Deep VAEs are notoriously challenging to stabilize and train. There are some recent, very impactful works that focus exclusively on training very deep VAEs by using regularization techniques such as gradient clipping or spectral regularization and combining effectively various implementation aspects (type of cells in the encoder/decoder, momentum of batch normalization, activation functions to be used), see [1], [2].
>
>   Specific to our model, we would like to highlight some implementation aspects without which we would not have managed to scale the model and achieve SOTA performance. In particular,
>
>   * **Naive implementation of attention**, where the keys and the queries are not direct outputs of the ResNet modules produced along with the context vector $\boldsymbol{c}^p_l$, but computed at a second stage, by a linear transformation of $\boldsymbol{c}^p_l$ implemented as 1x1 convolution, **would render the model too 'bulky' to train**.
>
>   * Similarly, the normalization scheme proposed in Equations 6, 7 is critical for scaling up the models. **Current attention normalization schemes** involve additional convolutions [3], which also **render deep VAEs difficult to train and had to be adapted**.
>
>   Therefore, we feel that **the architecture itself, albeit intuitively justified, has its own merit and it is not trivial to implement**.
>
>   We would also like to underscore the following **technical contributions of our work**:
>
>   1) The formulation of the **depth-wise attention mechanism, and its normalized version**, enables a very deep receptive field for a layer in the hierarchy. Naive 3D attention mechanisms to account for the 'third' dimension in the latent space, which is a byproduct of the factorization of the distributions and its organization in layers, would not be applicable in such architectures. As also suggested by Reviewer 3, it would be interesting to apply the proposed mechanism in ResNet architectures on different tasks, yielding attention driven dense connections (DenseNet) between the layers.
>
>   2) **Additional coordination between the prior and the posterior occurs when forming the conditioning factors of the posterior** (the generative network actively discovers the data features which complementary to the latent features yield the most descriptive context for the posterior). Novel works such as LadderVAE and NVAE keep the prior and the posterior in balance via their parametrization only (with a precision-weighted combination and with a residual correction  respectively).
>
>   We plan to restructure the paper in the revised version to better highlight these contributions.
>
>   [1] Rewon Child. Very deep vaes generalize autoregressive models and can outperform them on images.
>
>   [2] Arash Vahdat and Jan Kautz. Nvae: A deep hierarchical variational autoencoder.
>
>   [3] Chen M, Radford A, Child R, Wu J, Jun H, Luan D, Sutskever I. Generative pretraining from pixels.

---

> ### Comment · Reviewer_5Yd5 · 2021-11-28
> **Additional Discussion and Experiments Further improved the quality of the paper**
>
> Additional Discussion and Experiments Further improved the quality of the paper. Thanks to the authors to provide these insights. I believe this is a good paper that will add value to the VI community.

---

### Author Response · Authors · 2021-11-21
**Summary of Response to Reviews**

We thank all reviewers for their positive comments and thoughtful suggestions. We uploaded a significantly improved version of the manuscript which addresses all their suggestions and concerns. We respond to each one individually. Below, we summarize the changes in the revised submission.

A) In order to strengthen the empirical impact of our model, we have added four sets of experiments:

A.1 We have extended the qualitative assessment of the model. In Appendix D, we show that our model mitigates posterior collapse compared to current deep VAEs. In Appendix E, we visualize the attention maps.

A.2 In Appendix C, we provide additional ablation studies on our model.

A.3 In Table 11, we show that attentive VAE does not suffer from diminishing returns.

A.4 In Table 3, we push the performance of our model further, surpassing expensive auto-regressive models.


B) In order to improve the clarity and presentation of the main ideas introduced in the paper:

B.1 We describe the technical contributions of our work in a separate section (section 2.3). We have revised the presentation in sections 2.4, 2.5 to make use of the formulation in section 2.3.

B.2 We improved the quality of Figures 1, 2.

B.3 We clarified the cases in Table 5.

B.4 In Appendix A.3, we provide the detailed algorithms describing inference on the model.

B.5 Finally, We addressed other minor points. In particular, we i) improved the formatting of citations for consistency. ii) fixed typos iii) replaced terms global and local with holistic/ long-range and nearby/proximate respectively.

---

### Decision · Program_Chairs · 2022-01-20

**Decision:**

Accept (Poster)

**Comment:**

This paper adds an attention mechanism to deep variational autoencoders.  The authors develop a global + local attention method and achieve better log likelihoods than a variety of recent methods on MNIST and OMNIGLOT.  Overall the reviewers found this paper strong (8, 8, 8, 6), particularly after the author rebuttal.  They found the paper to be clear, the contribution sensible and novel and the experiments thorough and compelling.  In particular, the authors added additional experimental results on a larger dataset which addressed a common concern among the reviewers.  Thus the recommendation is to accept the paper.